# BRM-SWI/SNF chromatin remodeling complex enables functional telomeres by promoting co-expression of TRF2 and TRF1

**Shu Wu**[1◉], **Yuanlong Ge**[2◉], **Xiaocui Li**[1], **Yiding Yang**[1], **Haoxian Zhou**[1], **Kaixuan Lin**[3], **Zepeng Zhang**[2], **Yong Zhao**[1] *

**1** MOE Key Laboratory of Gene Function and Regulation, State Key Laboratory of Biocontrol, School of Life Sciences, Sun Yat-sen University, Guangzhou, China, **2** Key Laboratory of Regenerative Medicine of Ministry of Education, Institute of Aging and Regenerative Medicine, Jinan University, Guangzhou, China, **3** Yale Stem Cell Center & Department of Genetics, Yale School of Medicine, New Haven, Connecticut, United States of America

◉ These authors contributed equally to this work.
* zhaoy82@mail.sysu.edu.cn

**Data Availability Statement:** All relevant data are within the manuscript and its Supporting Information files.

## Abstract

TRF2 and TRF1 are a key component in shelterin complex that associates with telomeric DNA and protects chromosome ends. BRM is a core ATPase subunit of SWI/SNF chromatin remodeling complex. Whether and how BRM-SWI/SNF complex is engaged in chromatin end protection by telomeres is unknown. Here, we report that depletion of BRM does not affect heterochromatin state of telomeres, but results in telomere dysfunctional phenomena including telomere uncapping and replication defect. Mechanistically, expression of TRF2 and TRF1 is jointly regulated by BRM-SWI/SNF complex, which is localized to promoter region of both genes and facilitates their transcription. BRM-deficient cells bear increased TRF2-free or TRF1-free telomeres due to insufficient expression. Importantly, BRM deple-tion-induced telomere uncapping or replication defect can be rescued by compensatory expression of exogenous TRF2 or TRF1, respectively. Together, these results identify a new function of BRM-SWI/SNF complex in enabling functional telomeres for maintaining genome stability.

## Author summary

Human telomeres consist of repetitive "TTAGGG" DNA sequences and associated shel-terin complex, which maintain genomic stability by preventing linear chromosome ends from being recognized as broken DNA. TRF1 and TRF2, as key components of shelterin complex, directly associate with double strand telomeric DNA. In this study, we discov-ered that both TRF1 and TRF2 are jointly regulated by BRM-SWI/SNF complex. Deple-tion of BRM led to insufficient amount of TRF1 and TRF2, which is associated with telomere replication defect and telomere uncapping. More importantly, these phenomena can be rescued by ectopically expressed TRF1 and TRF2. Our work demonstrates a

**Funding:** This work was supported by the National Natural Science Foundation of China Grants (31970683, 31701196, 81702756), the National Key R&D Program of China (2018YFA0107000), and Guangzhou Municipal People's Livelihood Science and Technology Plan (201803010108 and 201604016111), Guangdong Basic and Applied Basic Research Foundation (2020A1515011522) and the Fundamental Research Funds for the Central Universities (18lgpy52). The funders had no role in study design, data collection and analysis, decision to publish, or preparation of the manuscript.

**Competing interests:** The authors have declared that no competing interests exist.

specific role of BRM-SWI/SNF complex on safeguarding genome stability by enabling functional telomeres.

## Introduction

Telomere is a specific DNA/protein structure localized at the end of linear chromosomes. In human cells, telomere is consisted with repetitive "TTAGGG" DNA sequences and telomere binding proteins termed shelterin complex, which prevents unnecessary degradation and activation of DNA damage response (DDR) at telomeres [1]. Shelterin complex comprises of six distinct proteins, disruption of shelterin complex or inhibition of its subunits induces telomere dysfunction, which can lead to either apoptosis or senescence [2].

As a key component of shelterin complex, TRF2 and TRF1 play an essential role by directly associating with double-stranded telomeric DNA, thus providing a vital platform for recruitment of additional shelterin proteins as well as non-shelterin factors crucial for the maintenance of telomere length and structure [2]. TRF2 is required to prevent ATM-mediated activation of DDR at telomeres likely through promoting the formation of t-loop, a structure that sequesters telomere terminus [3]. Loss of TRF2 results in telomere uncapping, leading to activation of DDR and accumulation of DNA damage factors at telomeres [4]. Ku70/80 are thus loaded onto open telomeres and initiate c-NHEJ, resulting in chromosome end to end fusion [5]. TRF1 mainly functions to facilitate the replication of telomeres, which pose a challenge for progression of replication fork due to the repetitive G-rich sequences of DNA [3]. Upon TRF1 deletion, cells display replication defect in telomeres and fragile telomeres that are characterized by multiple signals at single chromatid end [6].

Despite of the importance of shelterin proteins in end protection and replicative cell senescence, very little is known on the regulation of their expression. Prevailing paradigm is that shelterin protein is ubiquitously expressed to protect telomeres in all tissues. Recent studies challenge this view by showing that expression of shelterin components, notably TRF2 and TRF1, are strictly regulated. For instance, *in vivo* stoichiometry demonstrated that TRF2 and TRF1 are sufficiently abundant to cover all telomeric DNA [7]. While transcription factor Sp1 and β-catenin activate TRF2 transcription [8, 9], microRNA miR-23a and miR-155 suppresses TRF2 and TRF1 translation by targeting 3' UTR of their transcripts, respectively [10, 11].

The switch/sucrose nonfermentable (SWI/SNF) complexes belong to ATP-dependent chromatin remodeling complex, and have been conserved from yeast to humans. These complexes use the energy from ATP hydrolysis to remodel chromatin, impacting a variety of biological processes including gene transcription, DNA replication and DNA damage repair [12, 13]. In mammalian cells, SWI/SNF complexes are comprised of one of two mutually exclusive catalytic ATPase subunits BRM (SMARCA2) or BRG1 (SMARCA4) with a set of high conserved subunits (SNF5, BAF155 and BAF170), and other variant subunits [14]. Disruption of SWI/SNF function has been associated with tumorigenesis, as inactivating mutations in SWI/SNF subunits are often identified in a variety of cancer cells [13]. Previously, we revealed that BRG1-SWI/SNF chromatin remodeling complex is engaged in telomere length maintenance of human cancer cells by regulating hTERT expression [15]. Whether and how BRM-SWI/SNF complex plays a role in chromatin end protection is largely unknown.

In this study, we reported that depletion of BRM-SWI/SNF complex results in telomere dysfunction phenomena, including activation of ATM, appearance of telomere dysfunction induced foci (TIF), telomere replication defect and a rapid telomere loss and/or chromosome end to end fusion. Because BRM-SWI/SNF chromatin remodeling complex did not affect

heterochromatin state of telomeres, we suspected that BRM-SWI/SNF may regulate the expression of shelterin proteins. Indeed, it is revealed that BRM is recruited to the promoter of TRF2 and TRF1 and BRM depletion reduces mRNA and protein level of TRF2 and TRF1. Compensatory expression of exogenous TRF2 and TRF1 rescues dysfunctional telomeres and replication defect induced by BRM depletion. These results support that BRM-SWI/SNF remodeling complex is required to transcribe sufficient TRF2 and TRF1 for ensuring functional telomeres. BRM-SWI/SNF also represents a new mechanism by which one factor jointly regulates the expression of multiple genes with similar function.

## Results

### Genome instability and cell apoptosis induced by BRM depletion

To evaluate the function of SWI/SNF chromatin remodeling complex in maintaining genome stability, BRM, a key ATPase subunit in SWI/SNF complex, was depleted by siRNAs in immortalized fibroblast VA13 cells and human cervical cancer HeLa cells, representing ALT (Alternative lengthening of Telomeres) and telomerase positive cells, respectively (Fig 1A). Micronuclei are considered as a marker of genomic instability potentially induced by unrepaired DNA breaks that result in genomic remnants failing to segregate during cell division [16]. Our results showed that knockdown of BRM leads to significant increase of micronucleus in both VA13 and HeLa cells (Fig 1B–1D), indicating genome instability induced by BRM depletion. The same result was obtained when SaoS2, HepG2 and BJ fibroblast cells were used, indicating that observed phenomena are not cell type specific (S1 Fig).

In addition, we also observed that BRM-depleted cells display slowed or stopped proliferation (Fig 1E and 1F). Meanwhile, increased percentage of apoptotic cells was detected by FACS analysis in both BRM-depleted VA13 and HeLa cells (Fig 1G–1I). These results indicated that BRM-SWI/SNF complex is critical for maintenance of genome stability and for cell proliferation/survival.

### BRM deficiency induces dysfunctional telomeres

Telomere is a key element at the end of chromosome to safeguard the genome stability [1]. We thus investigated whether BRM deficiency affects the end protection function of telomeres. Because SWI/SNF complex acts to remodel chromatin structure, we suspected that depletion of BRM may change the state of heterochromatin, which is important for telomeres functioning to protect chromosome ends. Using micrococcal nuclease assay followed by hybridization with telomeric probe, we demonstrated no difference in size and distribution of nucleosome (Fig 2A), indicating that BRM-SWI/SNF complex may not safeguard genome stability through ensuring heterochromatin at telomeres, as another chromatin remodeling complex NoRC does [17].

However, we observed that BRM-depleted VA13 and HeLa cells displayed a significant increase in telomere dysfunction-induced foci (TIFs), which are manifested by γH2AX foci colocalizing with telomeres (Fig 2B–2E). Ataxia telangiectasia mutated (ATM) is a main sensor of DNA damage response (DDR), which responses to DNA double-stranded breaks and is activated by dysfunctional telomeres [18, 19]. In line with this, we observed activation of ATM (p-ATM) in BRM-deficient VA13 and HeLa cells (Fig 2F and 2G). These data supported the idea that BRM depletion results in dysfunctional telomeres that activates ATM-mediated DDR.

### BRM is required for telomeres capping and replication

To further explore how BRM deficiency affects chromosome end protection by telomeres, quantitative fluorescence in situ hybridization (q-FISH) was performed in BRM-depleted

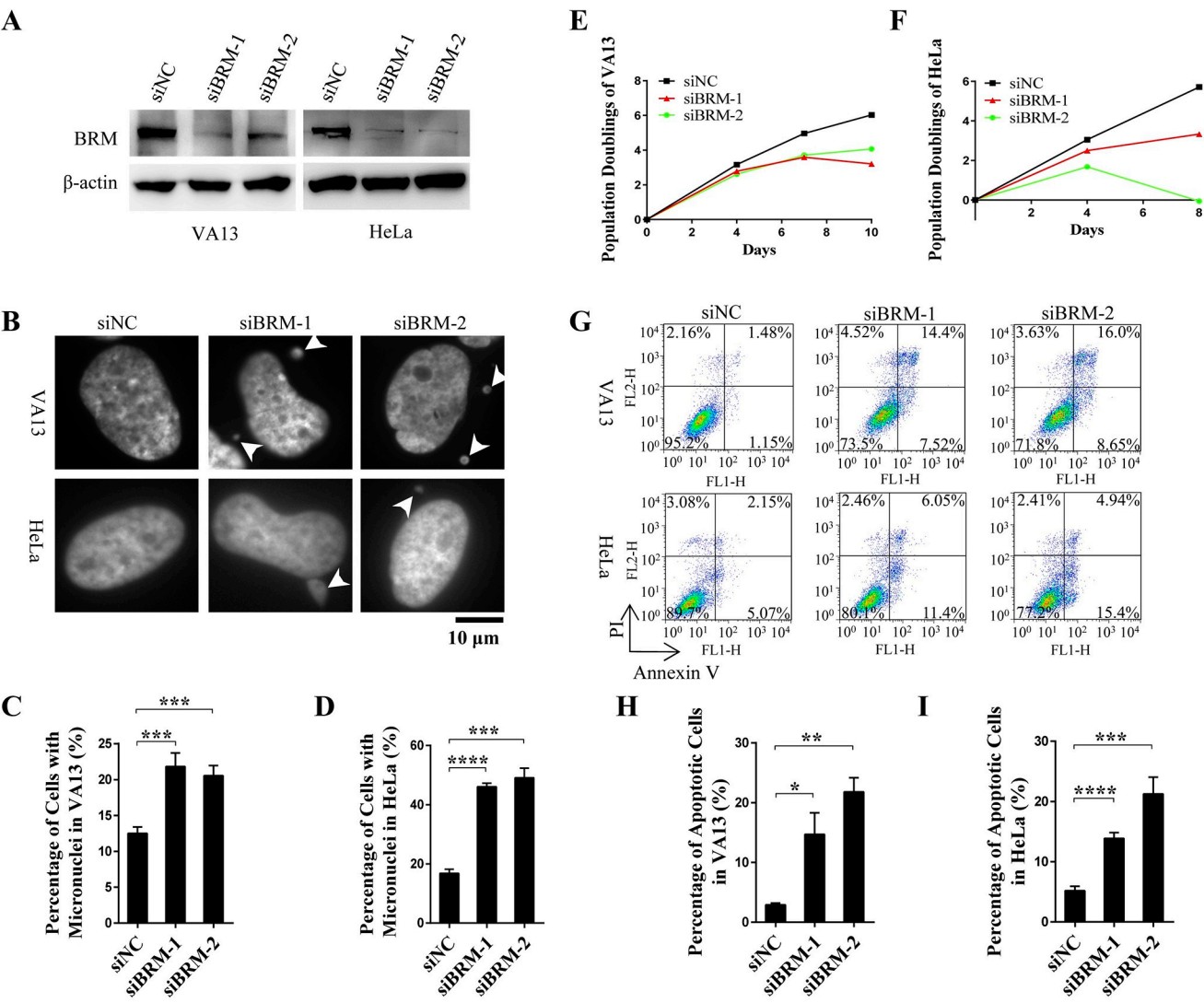

**Fig 1. BRM depletion induces genome instability and cell apoptosis.** (A) Western blot showing depletion of BRM in VA13 and HeLa cells by siRNA. (B) Detection of micronuclei in BRM-depleted VA13 and HeLa cells. (C) Quantification of the micronuclei of VA13 cells in (B), the fraction of cells with micronuclei were calculated. Data represent the mean ± SEM of three independent experiments (n ≥ 200 cells), ***P < 0.001. (D) Quantification of the micronuclei of HeLa cells in (B), the fraction of cells with micronuclei were calculated. Data represent the mean ± SEM of three independent experiments (n ≥ 100 cells), ***P < 0.001, ****P < 0.0001. (E) Proliferation of control and BRM-depleted cells in VA13 cells. (F) Proliferation of control and BRM-depleted cells in HeLa cells. (G) FACS analysis of apoptotic cells in control and BRM-depleted VA13 and HeLa cells. (H) Quantification of the apoptotic ratio of VA13 in (G). Data represent the mean ± SEM of three independent experiments, *P < 0.05, **P < 0.01. (I) Quantification of the apoptotic ratio of HeLa in (G). Data represent the mean ± SEM of three independent experiments, ***P < 0.001, ****P < 0.0001.

VA13 cells. While no change of telomere length upon BRM depletion was observed (Fig 3A and 3B), there was a significant increase of both telomere loss and chromosome end-end fusion in BRM depleted cells (Fig 3A, 3C and 3D). In together with appearance of TIF and ATM activation, these phenomena allowed us to conclude that BRM deficiency leads to telomere uncapping [2].

In addition, increased number of fragile telomeres was also observed in BRM-depleted cells (Fig 3A and 3E), indicating replication problem in telomeres. It has been reported that replication stress leads to accumulation of PCNA and RPA in stalled replication forks [20, 21], forming foci that can be detected by immunofluorescence (IF) and FISH. Indeed, we observed

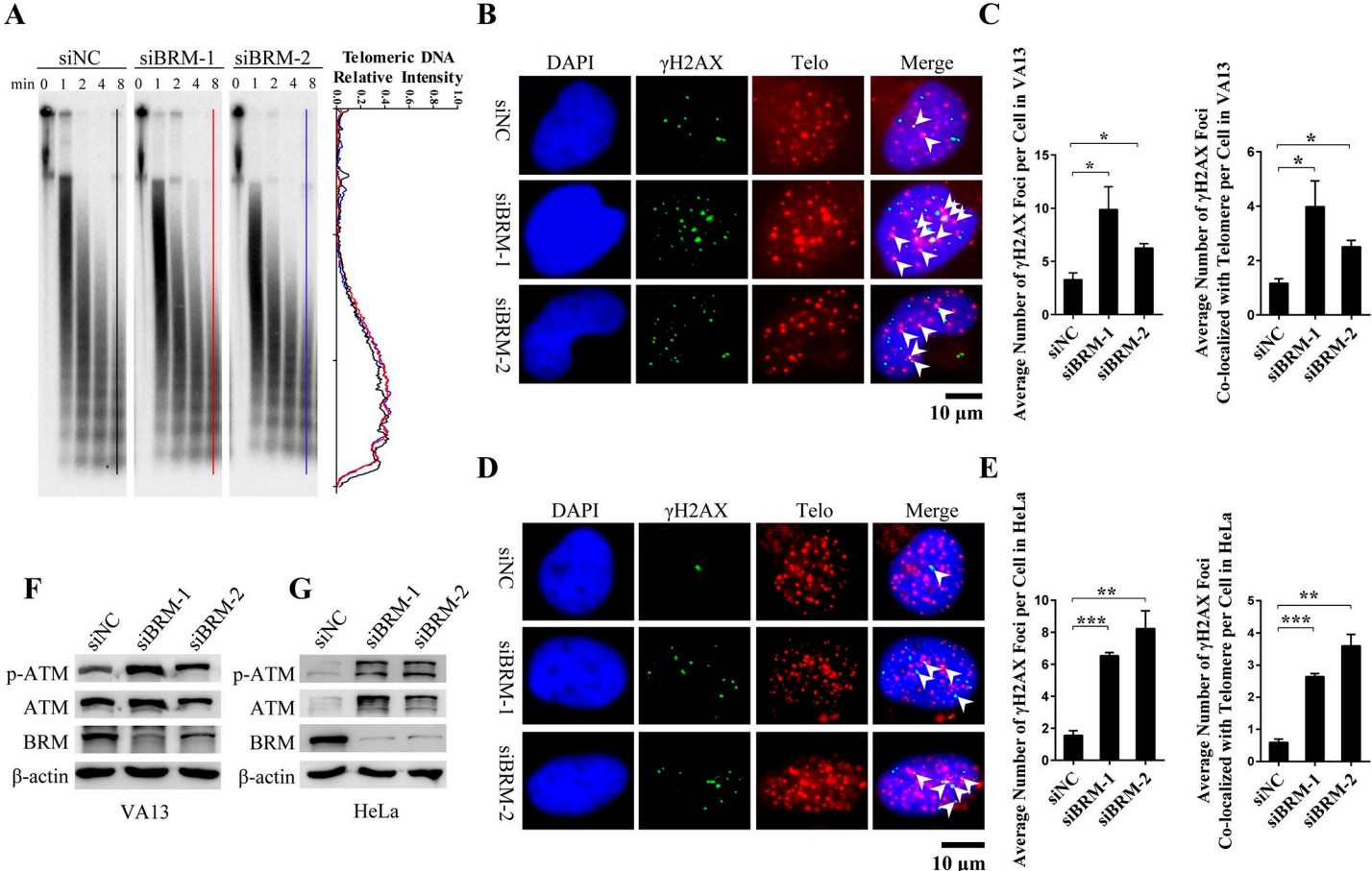

**Fig 2. Activation of DNA damage response at telomeres in BRM-depleted cells.** (A) Condensation of telomere chromatin was determined by micrococcal nuclease assay. (B) IF-FISH detection of γH2AX foci in control and BRM-depleted VA13 cells. (C) Quantification of (B). Data represent the mean ± SEM of three independent experiments (n ≥ 100 cells), *P < 0.05. (D) IF-FISH detection of γH2AX foci in control and BRM-depleted HeLa cells. (E) Quantification of (D). Data represent the mean ± SEM of three independent experiments (n ≥ 100 cells), **P < 0.01, ***P < 0.001. (F) Western blot showing ATM activation in BRM-depleted VA13 cells. (G) Western blot showing ATM activation in BRM-depleted HeLa cells.

significantly increased PCNA and RPA foci in BRM-depleted cells, which are mostly colocalized with telomeres (Fig 3F–3I). These results strongly suggested that BRM deficiency induces replication defect at telomeres.

## BRM deficiency leads to TRF2 and TRF1-free telomeres

Given the fact that telomere uncapping and telomere replication defect is highly associated with TRF2 and TRF1, respectively [6, 19, 22], and that BRM-SWI/SNF complex can regulate gene transcription by remodeling its local chromatin [12], it is speculated BRM may affect end protection and telomere replication by regulating TRF2 and TRF1 expression. To test this hypothesis, we examined a total amount of cellular TRF2/TRF1 and TRF2/TRF1 associating with telomeres in control and BRM-depleted cells. The result showed that depletion of BRM results in significant decrease of cellular TRF2 and TRF1 protein level (Fig 4A and 4E). In addition, the number of TRF2 and TRF1 foci in nucleus significantly decreased (Fig 4B, 4F, 4C and 4G). Accordingly, we observed that TRF2-free and TRF1-free telomeres have increased by nearly three folds in BRM-depleted cells as compared to control (Fig 4D and 4H). Consistently, TRF2 and TRF1-ChIP coupled with slot blot using telomeric probe revealed decreased

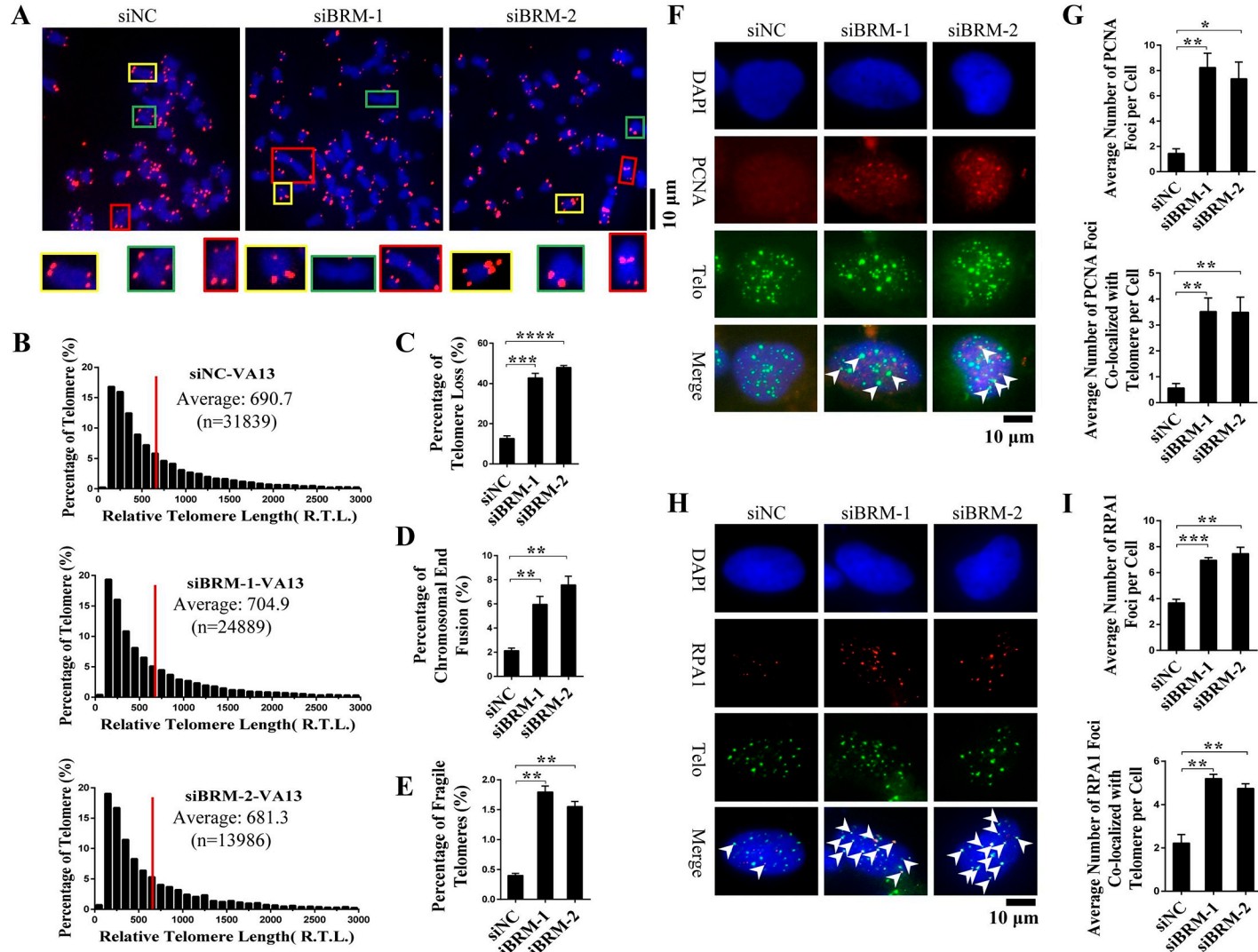

**Fig 3. Telomeres uncapping, telomere replication defect and fragile telomeres induced by BRM depletion.** (A) Metaphase telomere FISH detection of telomere loss and fusion or multiple telomeres signals at ends of chromosomes. Control or BRM-depleted VA13 cells were treated with nocodazole for 6 h, and subjected to FISH. (B) Q-FISH assay to determine telomere length of control and BRM-depleted VA13 cells. Number of telomeres analyzed (n) is indicated in figures, red bar indicated the average length of telomeres. (C) Quantification of (A). The percentage of chromosomes with one or more telomere-free ends was calculated. Data represent the mean ± SEM of three independent experiments, ***P < 0.001, ****P < 0.0001. (D) Quantification of (A). The percentage of chromosomes with fusion was calculated. Data represent the mean ± SEM of three independent experiments, **P < 0.001. (E) Quantification of (A). The percentage of fragile telomeres was calculated. Data represent the mean ± SEM of three independent experiments, **P < 0.001. (F) IF-FISH detection of PCNA foci in control and BRM-depleted VA13 cells. (G) Quantification of (F). Data represent the mean ± SEM of three independent experiments (n ≥ 100 cells), *P < 0.05, **P < 0.01. (H) IF-FISH detection of RPA1 foci in control and BRM-depleted VA13 cells. (I) Quantification of (H). Data represent the mean ± SEM of three independent experiments (n ≥ 100 cells), **P < 0.01, ***P < 0.001.

amount of telomeric DNA precipitated by TRF2 and TRF1, demonstrating decreased number of two proteins associating with telomeres (Fig 4I and 4J).

To test if genome instability observed in BRM-depleted cells is induced by TRF2 and TRF1 deficiency, TRF2 and TRF1 were knocked down in VA13 cells. Interestingly, we observed that depletion of TRF2 alone does not lead to increase of chromosome end fusion, however, concurrent depletion of TRF2 and TRF1 significantly increases the frequency of end fusion (S2 Fig), similar to that observed in BRM-depleted cells (Fig 3A and 3D).

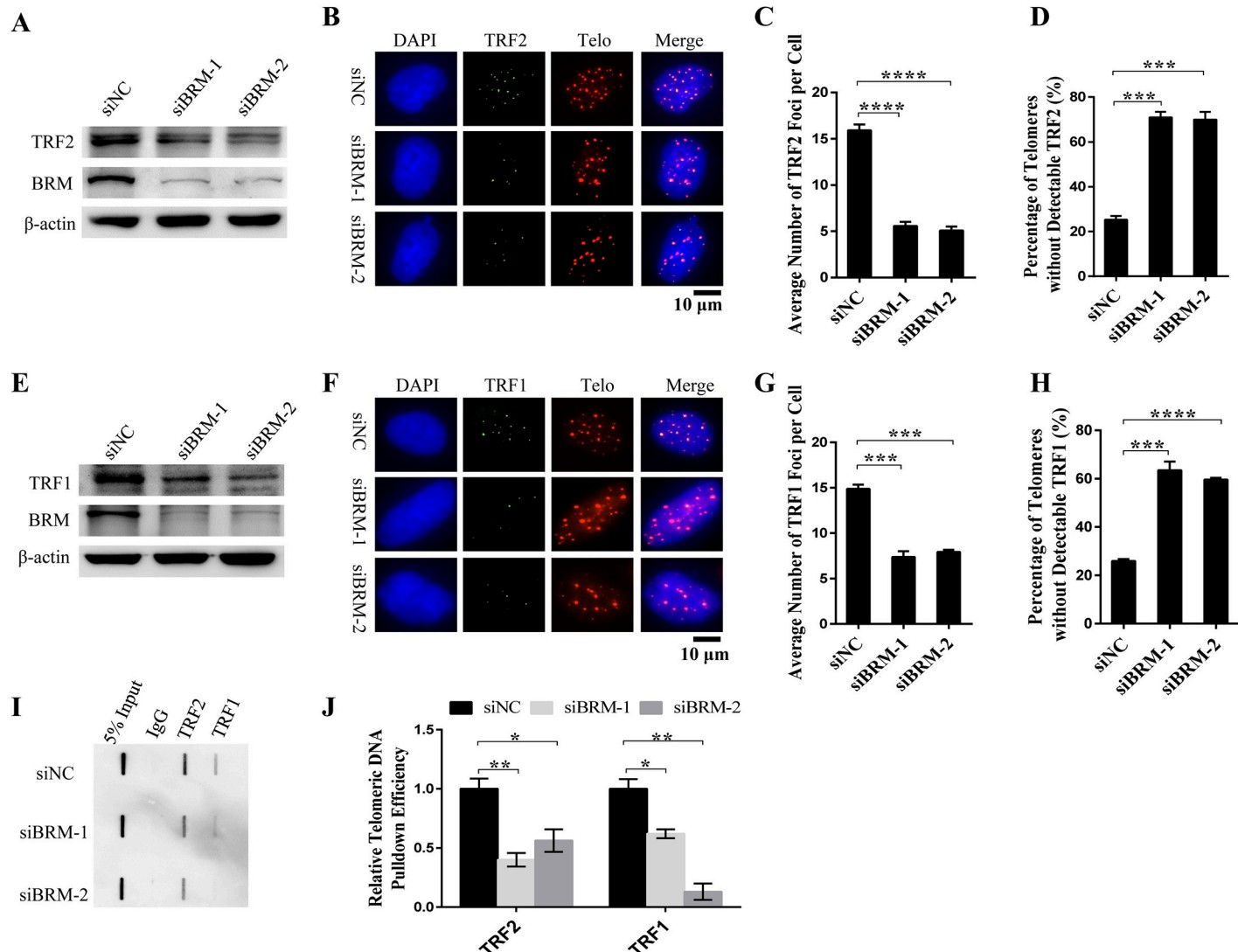

**Fig 4. BRM deficiency leads to TRF2 and TRF1-free telomeres.** (A) Western blot analysis of the protein level of TRF2 in control and BRM-depleted VA13 cells. (B) IF-FISH detection of TRF2 foci in control and BRM-depleted VA13 cells. (C) Quantification of (B). Average number of TRF2 foci in cells was calculated. Data represent the mean ± SEM of three independent experiments (n ≥ 100), ****P < 0.0001. (D) Quantification of (B). Percentage of telomeres lacking detectable TRF2 was calculated. Data represent the mean ± SEM of three independent experiments (n ≥ 100), ***P <0.001. (E) Western blot analysis of the protein level of TRF1 in control and BRM-depleted VA13 cells. (F) IF-FISH detection of TRF1 foci in control and BRM-depleted VA13 cells. (G) Quantification of (F). Average number of TRF1 foci in cells was calculated. Data represent the mean ± SEM of three independent experiments (n ≥ 100), ***P <0.001. (H) Quantification of (F). Percentage of telomeres lacking detectable TRF1 was calculated. Data represent the mean ± SEM of three independent experiments (n ≥ 100), ***P <0.001, ****P < 0.0001. (I) ChIP coupled with slot blot to determine telomeric DNA occupied by TRF2 and TRF1 in control and BRM-depleted VA13 cells. (J) Quantification of (I). Data represent the mean ± SEM of three independent experiments, *P < 0.05, **P < 0.01.

## BRM associates with TRF1 and TRF2 gene promoters and enhances their transcription

As a chromatin remodeling machinery, BRM-SWI/SNF complex associates with genome. To examine a potential association of BRM with TRF2 or TRF1 gene, we re-analyzed published whole-genome BRM ChIP-seq data [23]. Identified binding peaks and their locus on genome were shown (Fig 5A). We observed enrichment of BRM at the region of TRF2 and TRF1 promoters (Fig 5A), raising an intriguing possibility that BRM-SWI/SNF complex regulates TRF2

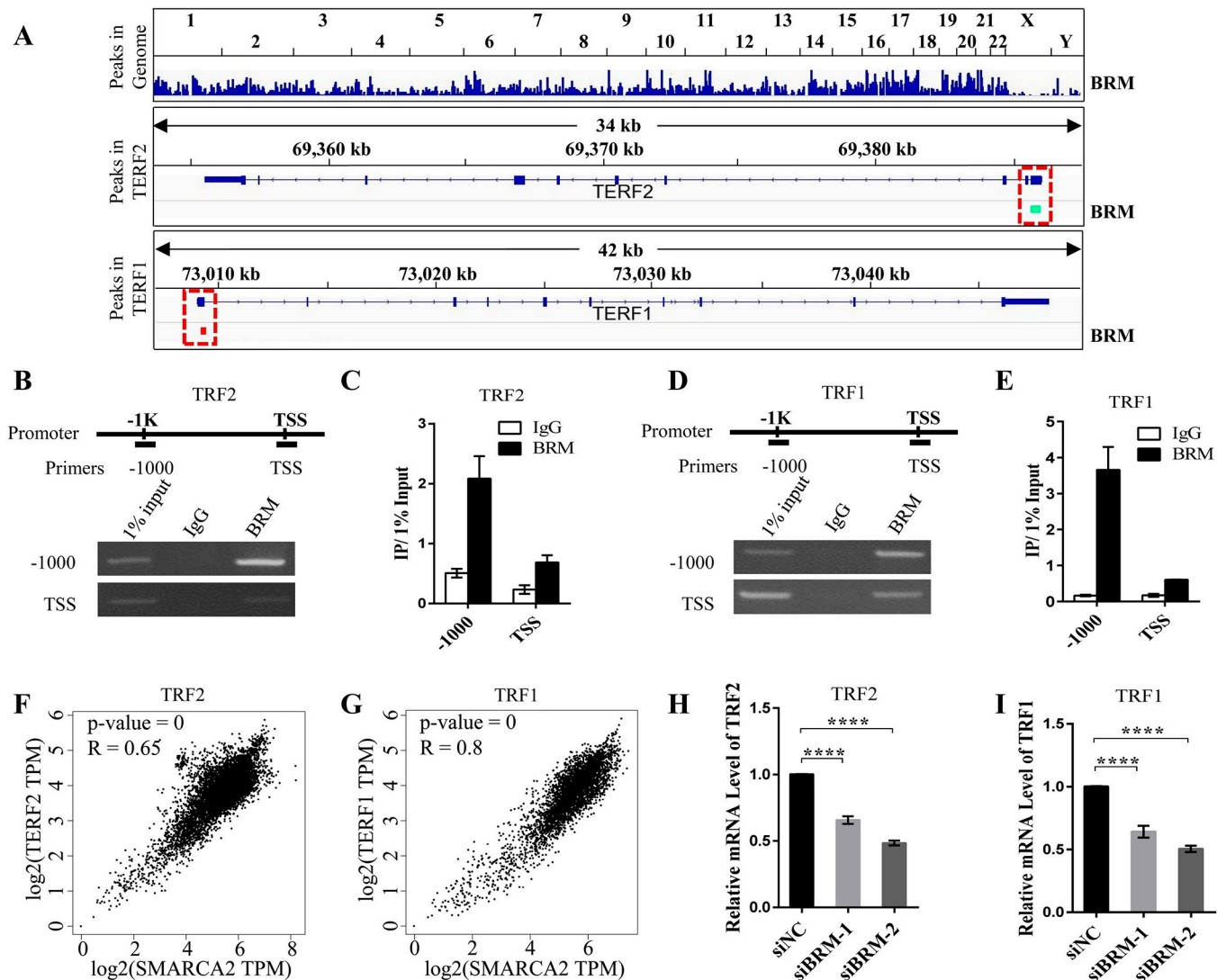

**Fig 5. BRM promotes TRF2 and TRF1 transcription by targeting their promoters.** (A) BRM occupancy of TRF2 and TRF1 promotor. Re-analysis of published ChIP-seq data illustrate that identified peaks enrichment at the promoter of TRF2 and TRF1 in HepG2 cells (upon shNS transfection[23]). Promoter region of interest was highlighted in red dotted box. (B) ChIP assay to determine the association between BRM with the promoter of TRF2. (C) Quantification of (B). Data represent the mean ± SEM of three independent experiments. (D) ChIP assay to determine the association between BRM with the promoter of TRF1. (E) Quantification of (D). Data represent the mean ± SEM of three independent experiments. (F) Gene expression correlation analysis between BRM (SMARCA2) and TRF2 (TERF2) using GEPIA with GTEx database. (G) Gene expression correlation analysis between BRM (SMARCA2) and TRF1 (TERF1) using GEPIA with GTEx database. (H) q-PCR detection of the mRNA level of TRF2 in control and BRM-depleted VA13 cells. Data represent the mean ± SEM of three independent experiments, ****P < 0.0001. (I) q-PCR detection of the mRNA level of TRF1 in control and BRM-depleted VA13 cells. Data represent the mean ± SEM of three independent experiments, ****P < 0.0001.

and TRF1 transcription by targeting and remodeling their promoters. We thus performed ChIP assay followed by PCR to demonstrate the interaction between BRM and TRF2 and TRF1's promoters. The results showed that BRM predominantly associates with DNA of TRF2 and TRF1 promoter that is 1 kb upstream of the transcription start site (TSS) (Fig 5B–5E).

Moreover, we also tested the correlation between BRM (SMARCA2) and TRF2 (TERF2) or TRF1 (TERF1) in expression using GEPIA (Gene Expression Profiling Interactive Analysis) and GTEx database [24]. By analyzing more than 1000 tissue samples, we found that TRF2 and TRF1 expression is positively correlated to level of BRM (SMARCA2) with p-value close to 0

and R value of 0.65 and 0.8, respectively (Fig 5F and 5G). Experimentally, we demonstrated that knockdown of BRM significantly reduced amount of TRF2 and TRF1 transcripts (Fig 5H and 5I). To confirm the generality of this finding, BRM regulating the expression of TRF2 and TRF1 had been examined in additional three cell lines (Saos2, HepG2, BJ fibroblast). We found that depletion of BRM significantly decreases mRNA level of TRF2 and TRF1 in all cell lines tested (S3 Fig).

In addition to TRF2 and TRF1, we speculated that BRM may also regulate the expression of other shelterin components. To this end, we examined the expression of POT1, RAP1, TPP1 and TIN2 upon BRM depletion. It was revealed that depletion of BRM leads to limited or no change of RAP1, TPP1 and TIN2 transcripts, but significant decrease of POT1. Western blot demonstrated unchanged protein level of RAP1 and TPP1, but increased POT1 (S4 Fig). Inconsistency between mRNA and protein level of POT1 may be due to translational and/or post-translational regulation that plays a critical role for metabolism of POT1.

### Rescue of telomere uncapping and replication defect by compensatory expression of TRF2 or TRF1

If BRM deficiency-induced telomere uncapping and replication defect is caused by lack of TRF2 and TRF1, it is speculated that compensatory expression of TRF2 or TRF1 in BRM-depleted cells would rescue the phenomena. To test it, TIF and telomeric PCNA and RPA foci were re-examined in BRM deficient cells expressing exogenous Flag-TRF2 or TRF1 (Fig 6A and 6B). Results showed that exogenous expression of TRF2 reduces the number of γH2AX foci and TIF to background level (Fig 6C and 6D). Similarly, exogenous expression of TRF1 reduced telomeric PCNA and RPA1 to the level similar to normal BRM wild-type cells (Fig 6E–6H). Therefore, BRM deficiency-induced telomere uncapping and replication defect can be rescued by exogenous expression of TRF2 or TRF1.

### Genome instability and cell apoptosis rescued by compensatory expression of TRF2 and TRF1

Rescued telomere capping and replication may re-ensure genome stability and prevent cells from undergoing apoptosis. To test it, micronuclei and cell apoptosis were re-examined in BRM-deficient cells expressing exogenous TRF2 and/or TRF1 (Fig 7A). Our results showed that compensatory expression of TRF2 or TRF1 alone can partially, but expression of both TRF2 and TRF1 simultaneously is able to completely rescue micronuclei formation and cell apoptosis induced by BRM depletion (Fig 7B–7E). These results indicate that genome instability and resulting apoptosis of BRM-depleted cells is mainly caused by telomere dysfunction induced by deficient TRF2 and TRF1.

### Discussion

For genes that are localized on persistent or temporary heterochromatin, chromatin status play an important role in transcription regulation, because the "closed" conformation of chromatin makes promoter region of genes generally inaccessible to transcription factors [25]. Transcriptionally active eukaryotic chromatin, which adopts an "open" conformation, is thus required for transcription factors and machinery to access cognate gene sequences [26]. Chromatin remodeling enzymes such as SWI/SNF complexes are able to alter chromatin status by promoting ATP-dependent change in chromatin structure [27]. SWI/SNF complexes are often recruited to enhancer or promoter region to regulate gene transcription [28, 29]. In this study, we showed that BRM-SWI/SNF complex is associated with both TRF2 and TRF1 promoter

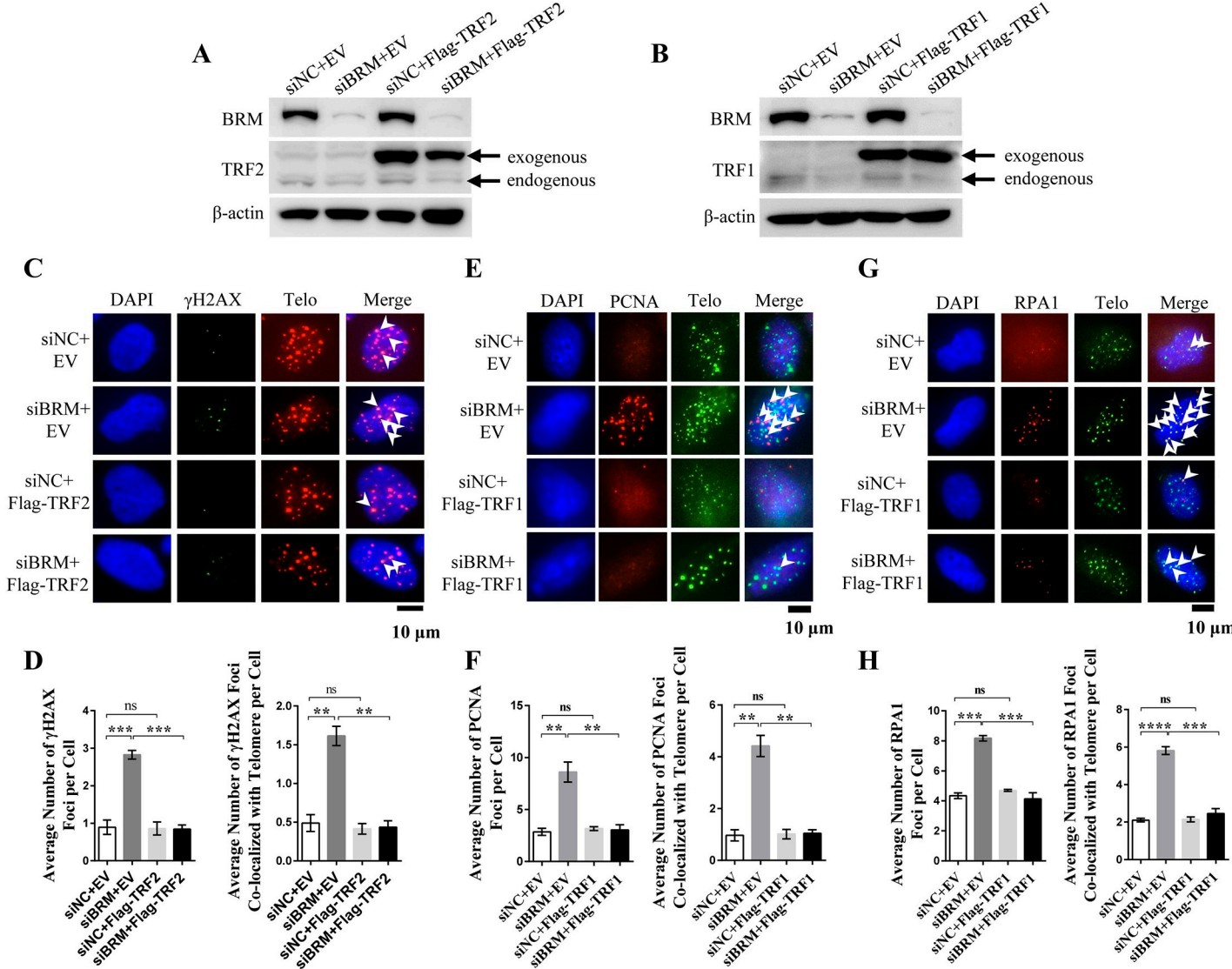

**Fig 6. Exogenous expression of TRF2 or TRF1 rescues BRM deficiency-induced telomere uncapping and replication defect.** (A) Western blot analysis of exogenous and endogenous TRF2 in BRM depleted cells. (B) Western blot analysis of exogenous and endogenous TRF1 in BRM depleted cells. (C) IF-FISH detection of γH2AX foci in control and BRM-depleted VA13 cells with EV or exogenous TRF2 expression. (D) Quantification of (C). Data represent the mean ± SEM of three independent experiments (n ≥ 100), **P < 0.01, ***P < 0.001. (E) IF-FISH detection of PCNA foci in control and BRM-depleted VA13 cells with EV or exogenous TRF1 expression. (F) Quantification of (E). Data represent the mean ± SEM of three independent experiments (n ≥ 100), **P < 0.01. (G) IF-FISH detection of RPA1 foci in control and BRM-depleted VA13 cells with EV or exogenous TRF1 expression. (H) Quantification of (G). Data represent the mean ± SEM of three independent experiments (n ≥ 100), ***P < 0.001, ****P < 0.0001.

and depletion of BRM leads to decrease of TRF2 and TRF1 transcripts (Fig 5). Therefore, BRM-SWI/SNF promotes TRF2 and TRF1 transcription likely through remodeling cognate chromatin structure to "open" conformation. Indeed, by analyzing ATAC-seq profile of TRF1 and TRF2 locus in control and BRM-depleted HAP1 cells (GEO accession: GSE108386) [30], we observed that depletion of BRM leads to reduction of chromatin accessibility at promoter region of TRF1 and TRF2 (S5 Fig).

Within shelterin complex, TRF2 and TRF1 is a concrete protein that directly associates with telomeric DNA, providing the platform for assembly of rest shelterin proteins and other non-shelterin factors [3]. It has been reported that TRF2 and TRF1 are among the most

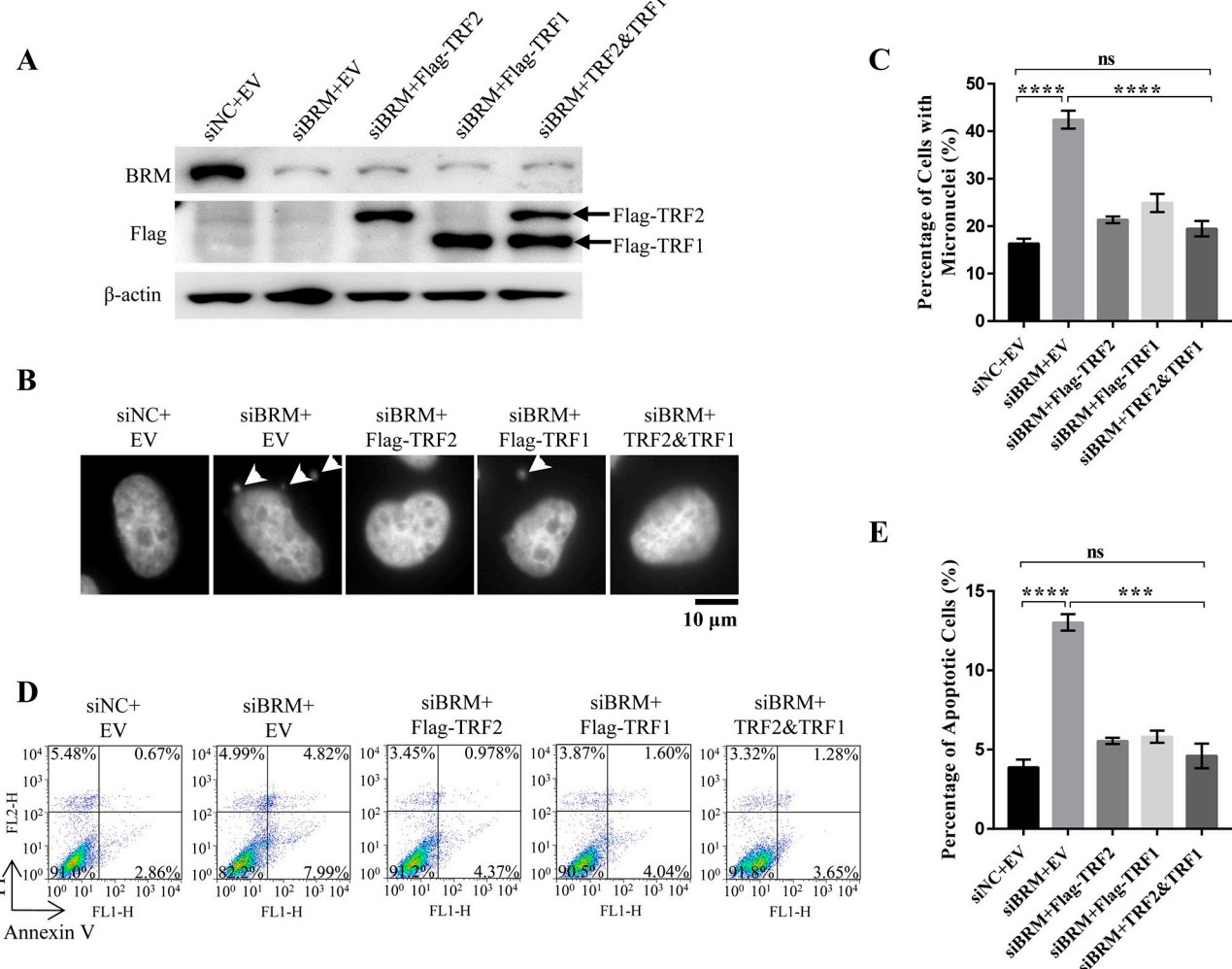

**Fig 7. Exogenous expression of TRF2 and TRF1 rescues BRM deficiency-induced genome instability and cell apoptosis.** (A) Western blot analysis of exogenous TRF2 and TRF1 expression and BRM knockdown efficiency. (B) Detection of micronuclei in control and BRM-depleted VA13 cells with EV or exogenous TRF2 or/and TRF1 expression. (C) Quantification of the micronuclei in (B), the fraction of cells with micronuclei were calculated. Data represent the mean ± SEM of three independent experiments (n ≥ 200 cells), ****P < 0.0001. (D) FACS analysis of apoptotic cells in control and BRM-depleted VA13 cells with EV or exogenous TRF2 or/and TRF1 expression. (E) Quantification of the apoptotic ratio in (D). Data represent the mean ± SEM of three independent experiments, ***P < 0.001, ****P < 0.0001.

abundant shelterin proteins in human normal and cancer cells [7]. Therefore, the mechanism is required to guarantee the substantial expression of TRF2 and TRF1. Our study showed that BRM-SWI/SNF chromatin remodeling complex is enriched in the promoter region of TRF2 and TRF1, which provides a basis for efficient transcription. This also raised an intriguing possibility that many other identified or unknown factors may coordinate with BRM-SWI/SNF to regulate the transcription of TRF2 and TRF1. Moreover, considering that TRF2 and TRF1 are equally important for end protection and that they exist at 1:1 stoichiometry in shelterin complex, the joint regulation of two gene's transcription by BRM-SWI/SNF would be the easiest way to modulate the amount of TRF2 and TRF1 in cells. Maintaining sufficient TRF2 and TRF1 is also important for proliferation and differentiation of embryonic stem (ES) cells [31–34]. When BRM is depleted in ES (H1) cells by siRNA, we observed mass death of cells (data not shown), indicating that BRM is an essential gene for ES cells survival. It is possible that

BRM dictates the fate of ES cells by regulating the expression of TRF2 and TRF1. This is worth further studying.

Mammalian SWI/SNF complexes contain one of two mutually exclusive, but structurally highly related catalytic ATPase subunits, BRM (SMARCA2) or BRG1 (SMARCA4) [14]. It is proposed that BRM and BRG1 may function in a complementary or antagonistic manner [35–37]. We found that promoters of TRF2 and TRF1 are not occupied by BRG1-SWI/SNF complex and that depletion of BRG1 does not affect the expression of TRF2 and TRF1 (S6 Fig). These results indicated that transcriptional regulation of TRF2 and TRF1 is a unique function of BRM.

Our study also supports previous finding that SWI/SNF complexes are vital to safeguard functional genome [38–40]. BRM-SWI/SNF remodeling complex has multiple functions including those involving DNA damage repair [29, 41]. It has been reported that BRM is required for recruitment of KU70/KU80 to DNA damage sites [42, 43]. In addition, BRM promotes the transcription of GTF2H1 that is indispensible for DNA damage repair by NER [29]. Although depletion of BRM may result in deficient DNA damage repair leading to genome instability, our results supported the idea that BRM depletion-induced genome instability is mainly caused by telomere dysfunction due to the lacking of TRF2 and TRF1, because exogenous expression of TRF2 and TRF1 completely rescues genome instability and cell apoptosis (Fig 7). This, however, does not exclude the possibility that BRM-SWI/SNF complex may also participate DNA damage repair at telomeres.

## Material and methods

### Cell culture and plasmids

VA13, HeLa, SaoS2, HepG2, BJ fibroblast, 293T were obtained from Chinese Academy of Sciences of Type Culture Collection and were cultured at 37˚C under 5% $CO_2$. Cells were grown in DMEM (Gibco) with 10% FBS (Gibco), 100 U/ml penicillin and 1% streptomycin (Gibco).

Full-length human cDNA of TRF2 and TRF1 were obtained from Dr. Zhou Songyang lab, and cloned into pLenti-Dest-EF1a-IRES-puro vector to generate plasmids Flag-TRF2 and Flag-TRF1.

### Gene silencing and overexpression

SiRNA was transfected into target cells using Lipofectamine RNAiMAX Transfection Reagent (Invitrogen), according to the manufacturer's instructions. siRNA against BRM (si-1:5'-GC UGAGAAACUGUCACCAAATdTdT-3'; si-2: 5'-GUCCUGGACCUCCAAGUGUCUdT dT-3'), TRF1 (5'-GGACAAGUGUCAUG UUAAAdTdT-3'), TRF2 (5'-ACAGAAGCAGU GGUCGAAUCdTdT-3'), BRG1 (si-1: 5'-ACAUGCACCAGAUGCACAAdTdT-3'; si-2: 5'-GG GUACCCUCAGGACAA CAdTdT-3') were provided by Suzhou GenePharma Co., Ltd. The scrambled sequence was used as a control. For TRF1, TRF2 stable expression cell lines, lentivirus was packaged in 293T cells using calcium phosphate transfection. Virus supernatants were collected and used to infect target cells. Empty vector was used as a control.

### Apoptosis assay

Cells were transfected with siRNA, after 3 days, cells were collected for apoptosis assay. Cells were co-stained with Annexin-V (FTIC) and PI (Annexin-V/PI apoptosis detection kit, KGA105, KeyGenBioTech) according to manufacturer's instructions. Stained cells were analyzed by FACS Calibur (BD Bioscience) and data were analyzed with FlowJo software.

## Micrococcal nuclease assay

Micrococcal nuclease assay was performed as previously described [44]. Briefly, $3 \times 10^6$ cells are needed. Cell pellet is suspended in 1 ml buffer A (100 mM NaCl, 10 mM Tris [pH 7.5], 3 mM MgCl$_2$, 1 mM CaCl$_2$, 0.5 mM phenylmethylsulfonyl fluoride, 1 × protease inhibitor cocktail), washed twice with buffer A, and then resuspended in 1 ml buffer A with 0.7% NP-40 to lyse cells. After gently mixing and incubating on ice for 5 min, nuclei were harvested at 2,000 rpm for 5 min and resuspended in 650 μl buffer A without NP-40. Aliquots of 100 μl were digested at 30°C with 5 U Micrococcal nuclease (NEB, M0247S) for 0, 1, 2, 4, 8 min. Reactions were stopped by adding 1 volume of TEES-protK (10 mM Tris-HCl [pH 7.5], 10 mM EDTA, 1% SDS, 50 μg/ml proteinase K) and incubating at 37°C for 2 h to overnight. DNA was extracted with phenol-chloroform, precipitated with isopropanol in the presence of 0.3 M sodium acetate [pH 5.2], and resuspended in 40 μl TE. Load 20 μl of each DNA sample and run in 1.2% agarose at 3 V/cm about 7 h until BPB shift to about 3 cm above the edge of the gel. The DNA was depurination with 0.2M HCl for about 5min until the BPB bands turn to yellow. Then resin the gel into the denature buffer (1.5 M NaCl, 0.5 M NaOH) for over 30 min. After neutralization (3 M NaCl, 0.5 M Tris-HCl [pH 7.0]) for 30 min, transferred DNA to a Hybond-N$^+$ membrane overnight. The DNA was UV-cross-linked to the membrane before hybridization with $^{32}$P-labeled telomere-specific probe.

## Re-analysis of public ChIP-seq data and ATAC-seq data

Published ChIP-seq data from HepG2 cells upon transfection with shNS (GEO accession: GSE102559) [23] was re-analyzed in this study. ChIP-seq raw data was obtained from the Sequence Read Archive repository (SRA, SRP115303) and uploaded to the Galaxy platform [45]. Reads were aligned to the human genome (hg38) with BWA (Galaxy Version 0.5.9), poor quality alignments and duplicates were subsequently filtered with SAMtools (Galaxy Version 1.9). To visualize ChIP-seq signal density, replicate datasets were merged with SAMtools and further processed using bamcoverage tool (Galaxy Version 3.3.0); resulting bigwig files were visualized using IGV genome browser [46]. Peaks were determined with MACS2 peak caller (Galaxy Version 2 2.1.1.20160309) [47] using the predicted function to estimate fragment size for all datasets and the following analysis parameters Minimum FDR (q-value) cutoff for peak detection 0.05. ATAC-seq data of wild-type and BRM knockout cells (GSE108386) [30] were downloaded from the Sequence Read Archive repository (SRA, SRP127341). Raw reads were mapped to hg19 genome using Bowtie 2 (v2.3.1) [48]. Signal tracks were generated by deepTools (v3.1.3)[49] and normalized as reads per genome coverage.

## Chromatin immunoprecipitation

Cells were cross-linked with 1% formaldehyde for 10 min at room temperature, reflection terminated by 1.25 mM Glycine, and washed twice with cold PBS, resuspended in SDS lysis buffer (50 mM Tris–HCl [pH 8.0], 10 mM EDTA, 1% SDS) and sonicated to fragments of 200 bp to 1 kb. The supernatant was pre-cleared with Protein-A/G agarose beads precoated with Escherichia coli genomic DNA. Chromatin immunoprecipitation (ChIP) was carried out overnight at 4°C with primary antibodies against BRM (11966, 1:100 dilution, Cell Signaling Technology), TRF1 (1:200 dilution, GTX77605, Genetex), TRF2 (1:200 dilution, 05–513, Merck) or IgG (Sangon). Beads were washed three times, and eluted with 0.1 M NaHCO$_3$ & 1% SDS, followed by reverse cross-linking and phenol-chloroform extraction. DNA fragments were precipitated by ethanol in the presence of NaAc and glycogen. PCR was carried out to identify DNA fragment enriched in complexes. The following primers were used to detect the fragments of TRF2 and TRF1 promoters: TRF2-TSS-forward: 5'-

ATTGCGGCCGGCACATCGGGAACTA-3'; TRF2-TSS-reverse: 5'-CGGCCATGATAGAA ACAGCGTTCCG-3'; TRF2-1000-forward: 5'-CACAGGTCCAGCATGGGATTCACAT-3'; TRF2-1000-reverse: 5'-TTGTCCTACCAGCCCCACTAGTCT-3'; TRF1-TSS-forward: 5'-GAGCCCTCGAATGCGAGCCAATCG-3'; TRF1-TSS-reverse: 5'-CTCGGGGCCGCTGAGG AAACATCC-3'; TRF1-1000-forward: 5'-GTGCATAAACGATGTTCAGTGAAT-3'; TRF1-1000-reverse: 5'-TTCAAGTTATCCTCCTGCCAAAGT-3'. Slot blot was used to detect TRF2 and TRF1 precipitated/occupied telomeric DNA with a biotin-labeled telomere-specific probe.

## Immunoblotting

Cells were directly lysed in 2 × SDS loading buffer and boiled for 15min. Proteins were separated by SDS–PAGE, transferred to PVDF membrane, and probed with antibodies specific for p-ATM (p-S1981, 1:5000 dilution, ab81292, Abcam), ATM (1:5000 dilution, ab32420, Abcam), TRF2 (1:2000 dilution, 05–513, Merck), TRF1 (1:1000 dilution, GTX77605, Genetex), POT1 (1:2000 dilution, NB500-176, Novus Biologicals), RAP1 (1:1000 dilution, 5433, Cell Signaling Technology), TPP1 (1:1000 dilution, 14667, Cell Signaling Technology), BRM (1:2000 dilution, ab15597, Abcam), BRG1 (1:2000 dilution, 21634-1-AP, Proteintech) and Flag (1:5000 dilution, F1804, Sigma). β-actin (1:5000 dilution, 66009-1-Ig, Proteintech) antibody was used as a loading control. HRP-conjugated anti-rabbit or anti-mouse (KPL, Inc) were then used.

## Quantitative real-time PCR

Total RNA was extracted from cells using RNAiso Plus Reagent (9109, Takara) according to manufacturer's instructions. 1.0 μg of total RNA was reverse-transcribed to cDNA using PrimeScript RT reagent Kit (RR047A, Takara). cDNA was used for real-time PCR using 2 × RealStar Green Fast Mixture (A311-10, GenStar). β-actin was used as internal control for all experiments. The following primers were used for amplification: β-actin-forward: 5'-CATG TACGTTGCTATCCAGGC-3'; β-actin-reverse: 5'-CTCCTTAATGTCACGCACGAT-3'; TRF2-forward: 5'-GTACGGGGACTTCAGACAGAT-3'; TRF2-reverse: 5'-CGCGACAGA-CACT GCATAAC-3'; TRF1-forward: 5'-AACAGCGCAGAGGCTATTATTC-3'; TRF1-reverse: 5'-CCAAGGGTGTAATTCGTTCATCA-3'; POT1-forward: 5'-CAGCCAATGCAGTA TTTTGACC-3'; POT1-reverse: 5'-GGTGCCATCCC ATACCTTTAGAA-3'; RAP1-forward: 5'-GCGTCTGGTCAGAGAGCTG-3'; RAP1-reverse: 5'-TCAATCCTCCGAGCTACATTC T-3'; TPP1-forward: 5'-CCTCCACACGGTGCAAAAATG-3'; TPP1-reverse: 5'-CTCTGCTT GTCGG ATGCTCAG-3'; TIN2-forward: 5'-ACGCCTTTGTATGGGCCTAAA-3'; TIN2-reverse: 5'-AAGTTTCCTGTGCCTCCAAAAT-3'; BRM-forward: 5'-AGCGGGAATACA GACTTCAGG-3'; BRM-reverse: 5'-AAGTGCTTTTAGTTCCACGGTT-3'; BRG1-forward: 5'-AATGCCAAGCAAGATGTCGAT-3'; BRG1-reverse: 5'-AATGCCAAGCAAGATGTC GAT-3'.

## IF-FISH

Briefly, cells on the coverslip were fixed with 4% paraformaldehyde, then permeabilized with 0.5% Triton X-100 (in 1 × PBS). Cells were incubated overnight at 4˚C with primary antibodies against γH2AX (1:200 dilution, 2577, Cell Signaling Technology), PCNA (1:200 dilution, GTX100539, Genetex), RPA1 (1:200 dilution, sc-28304, Santa Cruz), TRF2 (1:200 dilution, 05–513, Merck), or TRF1 (1:50 dilution, GTX77605, Genetex), washed three times with 1 × PBST, and incubated with secondary antibodies (1:2000 dilution, DyLight488-conjugated anti-mouse or anti-rabbit, KPL). Cells were washed three times with 1 × PBST, re-fixed with 4% paraformaldehyde for 30 minutes, dehydrated by ethanol series solution, denatured at 85˚C for 5 minutes, and then hybridized with Cy3-labeled (CCCTAA)$_3$ PNA probe (Panagene)

at 37˚C for 4 hours. The cells were washed and mounted with DAPI. Fluorescence was detected and imaged using Zeiss Axion Imager Z1 microscope.

## Q-FISH

Cells were incubated with 0.5 μg/ml nocodazole (sigma) for 6 h to enrich cells at metaphases. Metaphase-enriched cells were hypotonic treated with 75 mM KCl solution, fixed with methanolglacial acetic acid (3:1), and spread onto clean slides. Telomeres were denatured at 85˚C for 4 min and hybridized with Cy3-labeled $(CCCTAA)_3$ PNA probe (Panagene). Chromosomes were stained with DAPI. Fluorescence from chromosomes and telomeres was digitally imaged on a Zeiss Axion Imager Z1 microscope. For quantitative measurement of telomere length, telomere fluorescence intensity was integrated using the TFL-TELO program.

## Statistical analysis

Statistical analysis was performed using GraphPad Prism version 7 (GraphPad Software). Data are expressed as the mean ± SEM. Comparisons between groups were made using a 2-tailed unpaired Student's t test. A P value of less than 0.05 was considered statistically significant.

## Supporting information

**S1 Fig. BRM depletion leads to significant increase of micronucleus in SaoS2, HepG2 and BJ cells.** (A) Western blot showing depletion of BRM in SaoS2, HepG2 and BJ cells by siRNA. (B) Detection of micronuclei in BRM-depleted SaoS2, HepG2 and BJ cells. (C)-(E) Quantification of the micronuclei of SaoS2, HepG2 and BJ cells in (B), the fraction of cells with micronuclei were calculated. Data represent the mean ± SEM of three independent experiments (n ≥ 100 cells), $^{**}P < 0.01$, $^{***}P < 0.001$.
(TIF)

**S2 Fig. Depletion of both TRF1 and TRF2 induces chromosome end fusion.** (A) Western blot showing depletion of TRF1 and TRF2 in VA13 cells by siRNA. (B) Metaphase telomere FISH detection of chromosome end fusion. Control or TRF1/TRF2-depleted VA13 cells were treated with colchicine for 6 h, and subjected to FISH. (C) Quantification of (B). Data represent the mean ± SEM of three independent experiments, $^{*}P<0.01$.
(TIF)

**S3 Fig. Depletion of BRM decreases the mRNA level of TRF2 and TRF1 in HeLa, SaoS2, HepG2 and BJ cells.** (A)-(C) q-PCR determination of the level of TRF2, TRF1 and BRM in HeLa cells transfected with siRNAs. (D)-(F) q-PCR determination of the level of TRF2, TRF1 and BRM in SaoS2 cells transfected with siRNAs. (G)-(I) q-PCR determination of the level of TRF2, TRF1 and BRM in HepG2 cells transfected with siRNAs. (J)-(L) q-PCR determination of the level of TRF2, TRF1 and BRM in BJ cells transfected with siRNAs. Data represent the mean ± SEM of three independent experiments, $^{*}P<0.05$, $^{**}P <0.01$, $^{***}P<0.001$, $^{****}P < 0.0001$.
(TIF)

**S4 Fig. Expression regulation of POT1, RAP1, TPP1 and TIN2 by BRM.** (A) q-PCR determination of mRNA level of POT1, RAP1, TPP1 and TIN2 in control and BRM-depleted VA13 cells. Data represent the mean ± SEM of three independent experiments, $^{*}P<0.05$, $^{**}P < 0.01$, $^{***}P<0.001$. (B) Western blot showing the protein level of TRF1, TRF2, POT1, RAP1 and TPP1 in control and BRM-depleted VA13 cells.
(TIF)

**S5 Fig. Assay for transposase-accessible chromatin sequencing (ATAC-seq) of TRF1 and TRF2 locus in control and BRM-depleted HAP1 cells.** Data are from GEO accession: GSE108386.
(TIF)

**S6 Fig. BRG1 does not regulate the expression of TRF2 and TRF1.** (A) Re-analysis of TRF2 and TRF1 genes occupied by BRG1. Data are from published BRG1 ChIP-seq in HepG2 cells. (B) Western blot showing depletion of BRG1 in VA13 cells by siRNAs. (C) q-PCR detection of the mRNA level of TRF2 in control and BRG1-depleted VA13 cells. Data represent the mean ± SEM of three independent experiments. (D) q-PCR detection of the mRNA level of TRF1 in control and BRG1-depleted VA13 cells. Data represent the mean ± SEM of three independent experiments.
(TIF)

## Acknowledgments

We thank all the members in Dr. Zhao's laboratory for insightful scientific discussion. We thank Dr. Zhou Songyang at the School of Life Sciences, Sun Yat-sen University for providing full-length human cDNA of TRF2 and TRF1.

## Author Contributions

**Conceptualization:** Shu Wu, Yong Zhao.

**Data curation:** Shu Wu, Yuanlong Ge, Xiaocui Li, Yiding Yang, Haoxian Zhou, Kaixuan Lin, Zepeng Zhang, Yong Zhao.

**Formal analysis:** Shu Wu, Yuanlong Ge, Xiaocui Li, Haoxian Zhou, Kaixuan Lin, Zepeng Zhang.

**Funding acquisition:** Shu Wu, Yuanlong Ge, Yong Zhao.

**Investigation:** Shu Wu, Yuanlong Ge, Xiaocui Li, Yiding Yang, Yong Zhao.

**Methodology:** Shu Wu, Yuanlong Ge, Xiaocui Li, Yiding Yang.

**Project administration:** Shu Wu, Yuanlong Ge, Yong Zhao.

**Resources:** Shu Wu, Yuanlong Ge, Yong Zhao.

**Validation:** Shu Wu.

**Visualization:** Shu Wu, Yuanlong Ge, Xiaocui Li.

**Writing – original draft:** Shu Wu, Yong Zhao.

**Writing – review & editing:** Shu Wu, Yuanlong Ge, Yong Zhao.

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
