## [Decision Letter · Decision Letter 0]

12 Feb 2020

Dear Dr Zhao,

Thank you very much for submitting your Research Article entitled 'BRM-SWI/SNF Chromatin Remodeling Complex Enables Functional Telomeres by Promoting Co-expression of TRF2 and TRF 1' to PLOS Genetics. Your manuscript was fully evaluated at the editorial level and by independent peer reviewers. The reviewers appreciated the attention to an important problem, but raised some substantial concerns about the current manuscript. Based on the reviews, we will not be able to accept this version of the manuscript, but we would be willing to review again a much-revised version. We cannot, of course, promise publication at that time.

If you decide to revise the manuscript for further consideration at PLOS Genetics, please aim to resubmit within the next 60 days, unless it will take extra time to address the concerns of the reviewers, in which case we would appreciate an expected resubmission date by email to plosgenetics@plos.org.

[LINK]

We are sorry that we cannot be more positive about your manuscript at this stage. Please do not hesitate to contact us if you have any concerns or questions.

Yours sincerely,

Jin-Qiu Zhou

Associate Editor

PLOS Genetics

Gregory Barsh

Editor-in-Chief

PLOS Genetics

Reviewer's Responses to Questions

**Comments to the Authors:**

Reviewer #1: This manuscript described the molecular basis for how BRM-SWI/SNF complex contributes to telomere maintenance. The authors reported that RMB associates with TRF1 and TRF2 promoters and enhances their transcription. BRM knockdown led to reduced TRF1 and TRF2 expression and their occupancy at telomeres. In addition, BRM knockdown caused telomere uncapping, measured by gamma-H2AX foci formation at telomeres and chromosome end-to-end fusions; and telomere replication defects, evident by telomeric PCNA and RPA foci formation and fragile telomeres, which were mitigated by overexpressing TRF1 and TRF2. The work described in this manuscript is novel. The manuscript is well written. The experiments were well designed.

Major comments:

1. The authors demonstrated that some telomeres are TRF1 and TRF2 free in BRM knockdown cells by IF and FISH experiments. Did the authors quantify telomeric occupancy of TRF1 and TRF2 by ChIPs? Besides TRF1 and TRF2, did BRM deficiency affect other shelterin gene expression? Dr. de Lange laboratory showed that deletion in both TRF1 and TRF2 resulted in shelterin-free telomeres in mice. So Are TRF1- and TRF2-free telomeres bound by other shelterin proteins? It would be interesting to explore these questions in the follow-up studies, if these experiments are outside the scope of what could be accomplished within a timely revision period.

2. TRF1 and TRF2 overexpression could cause telomere and genome abnormalities, including rapid telomere loss, anaphase bridge, et al. Thus, it is important to show endogenous TRF1 and TRF2 expression in Figure 6A and B to justify TRF1 and TRF2 overexpression levels.

3. BRM knockdown cells showed increased end-to-end fusion. These cells had about 50% reduction in TRF2 levels (Figure 4). Because end-to-end fusion occurs when most TRF2 molecules are depleted in human cancer cell lines, I wonder are there other mechanisms contributing to the end-to-end fusion phenotype in BRM knockdown cells?

Minor comments:

Figure 5. TS should be TSS.

Page 5-6. “…BRM-SWI/SNF complex plays a role in chromatin end protection by telomeres…”. “By telomeres” is confusing.

TIF analysis: Telomere signals are poor. How the authors define gamma-H2AX at telomeric site?

Reviewer #2: In this manuscript the authors define an essential role for BRM in maintaining telomere function by regulating the expression of TRF1 and TRF2. The experiments are logical and well-done, and the results are convincing that this mechanism is relevant in the two cell lines tested. The major concern is the generality of the phenotype across all cell types. While the correlation between BRM and TRF1 and TRF2 across cell lines is compelling, testing the dependency in additional cell lines would be essential to really support the conclusion. Testing BRM knockdown across a panel of cell lines for a dependency for TRF1/TRF2 transcriptional regulation along with at least one of the assays for genome stability would be sufficient.

Is this a unique function for BRM that cannot be performed by BRG1? While it is clear that these two paralogs have functional differences, the ChIP-Seq data from HepG2 cells used in this manuscript (Raab et al 2017) indicates that BRM sites are also bound by BRG1 and that knockdown of each subunit can (but not necessarily) regulate genes in a similar manner. Does BRG1 knockdown similarly affect TRF1 and TRF2 expression or is this function exclusive to BRM? Based on a quick search, BRG1 expression displays the same correlation with TRF2 or TRF1 in the GTEx database. Why focus on BRM and not BRG1 for this study?

In a related issue, the BRM knockout mouse is reported to be phenotypically normal (although issues with whether or not these mice are actually knockouts have been raised). Could this mean that telomere dysfunction would only be observed upon acute depletion and that BRG1 can compensate in a long-term knockout setting? Can BRM knockout cells be cultured long term? Since many groups are trying to develop ways to specifically target BRM in BRG1 mutant lung cancers, understanding long term ramifications of BRM deletion in cells expressing both BRG1 and BRM is particularly important.

Reviewer #3: Title: BRM-SWI/SNF Chromatin Remodeling Complex Enables Functional Telomeres by Promoting Co-expression of TRF2 and TRF 1

This paper shows that BRM-SWI/SNF Chromatin Remodeling Complex is important for maintaining telomere function by promoting expression of TRF1 and TRF2. BRM-SWI/SNF complex remodels chromatin and binds to TRF2 and TRF1 promoters to regulate their expression. In contrast, depletion of BRM decreases TRF2 and TRF1

transcripts. The findings are novel and interesting to the field of telomere biology. Excitingly, BRM depletion-induced genome instability is mainly caused by telomere dysfunction due to the lacking of TRF2 and TRF1, and exogenous expression of TRF2 and TRF1 completely rescues genome instability and cell apoptosis. Previously,

the authors revealed that BRG1-SWI/SNF chromatin remodeling complex is engaged in

telomere length maintenance by regulating hTERT expression (15). Does BRG1 SWI/SNF also regulates expression of TERT or TRF1/2? Is BRG1 or BRM function in telomere cell type specific?

Specific comments:

1) VA13, HeLa, 293T were used for this study. Why these three cell lines were used? The reasons were not given.

2) Chromatin remodeling complex and telomere regulation are important for embryonic stem cells, so embryonic stem cells could be good model study the function of BRM-SWI/SNF Chromatin Remodeling Complex and TRF1 and TRF2. Does BRM-SWI/SNF regulate TRF2 and TRF1 in embryonic stem cells? or discussed?

3) Re-analysis of public ChIP-seq data

Published ChIP-seq data from HepG2 cells upon transfection with shNS (GEO accession: GSE102559) (23) was re-analyzed in this study. The biding of the protein by ChIP may differ among cell types. Can the ChIP biding data from HepG2 cells be applied to other three cell types used in this study?

4) The authors imply that BRM-SWI/SNF remodels chromatin and activates TRF2 and TRF1 expression. But no data is available to show that BRM-SWI/SNF open chromatin at TRF2 and TRF1 promoters or enhancers. It is unknown whether BRM-SWI/SNF opens chromatin at genome-wide. Perhaps ATAC may provide some answers.

**Have all data underlying the figures and results presented in the manuscript been provided?**

Reviewer #1: Yes

Reviewer #2: Yes

Reviewer #3: Yes

PLOS authors have the option to publish the peer review history of their article (what does this mean?). If published, this will include your full peer review and any attached files.

Reviewer #1: No

Reviewer #2: No

Reviewer #3: No

---

## [Decision Letter · Decision Letter 1]

26 Apr 2020

Dear Dr Zhao,

We are pleased to inform you that your manuscript entitled "BRM-SWI/SNF Chromatin Remodeling Complex Enables Functional Telomeres by Promoting Co-expression of TRF2 and TRF1" has been editorially accepted for publication in PLOS Genetics. Congratulations!

Yours sincerely,

Jin-Qiu Zhou

Associate Editor

PLOS Genetics

Gregory Barsh

Editor-in-Chief

PLOS Genetics

Comments from the reviewers (if applicable):

Reviewer's Responses to Questions

**Comments to the Authors:**

Reviewer #1: the authors have addressed all my questions.

Reviewer #2: The authors did an outstanding job of addressing concerns. The manuscript is significantly improved.

Reviewer #3: The authors have addressed my concerns by additional experiments and data analysis. No more comments.

**Have all data underlying the figures and results presented in the manuscript been provided?**

Reviewer #1: Yes

Reviewer #2: Yes

Reviewer #3: Yes

PLOS authors have the option to publish the peer review history of their article (what does this mean?). If published, this will include your full peer review and any attached files.

Reviewer #1: No

Reviewer #2: No

Reviewer #3: Yes: L Liu

**Data Deposition**

http://datadryad.org/submit?journalID=pgenetics&manu=PGENETICS-D-19-02053R1

**Press Queries**

---

## [Editor Report · Acceptance letter]

29 May 2020

PGENETICS-D-19-02053R1 

BRM-SWI/SNF Chromatin Remodeling Complex Enables Functional Telomeres by Promoting Co-expression of TRF2 and TRF1 

Dear Dr Zhao, 

We are pleased to inform you that your manuscript entitled "BRM-SWI/SNF Chromatin Remodeling Complex Enables Functional Telomeres by Promoting Co-expression of TRF2 and TRF1" has been formally accepted for publication in PLOS Genetics! Your manuscript is now with our production department and you will be notified of the publication date in due course.

With kind regards,

Jason Norris

PLOS Genetics

On behalf of:
